# Purification and Identification of EPS Produced by Five Lactic Acid Bacteria and Evaluation of Their Effects on the Texture of Fermented Goat Milk

Gengdian Liu [1], Guowei Shu [1,*], Jiansheng Wang [2], Zhanmin Wang [3], Yu Liu [4], Yilin Li [1] and Li Chen [5,*]

[1] School of Food Science and Engineering, Shaanxi University of Science and Technology, Xi'an 710021, China; gengdiancsy@163.com (G.L.); mingzhi202138@163.com (Y.L.)
[2] Shaanxi Hongxing Meiling Dairy Co., Ltd., Weinan 711799, China; wangjshhx@gmail.com
[3] Fuping County Milkgoat Dairy Co., Ltd., Weinan 711711, China; zhanminmg@gmail.com
[4] Shaanxi Heshi Dairy Co., Ltd., Baoji 721200, China; liuyuhsh@gmail.com
[5] College of Food Engineering and Nutritional Science, Shaanxi Normal University, Xi'an 710119, China
* Correspondence: shuguowei@gmail.com (G.S.); chenlisp@snnu.edu.cn (L.C.)

**Abstract:** Extracellular polysaccharide (EPS) produced by five lactic acid bacteria (Limosilactobacillus fermentum B55, Limosilactobacillus fermentum B62, Lactiplantibacillus plantarum 7830, Pediococcus acidilactici B30, and Lactobacillus helveticus K2) were purified and identified, and their effects on the texture of fermented goat milk were evaluated. The purified EPS fractions EPS 1a, EPS 2b, EPS 3c, EPS 4d, and EPS 5e of strains B62, 7830, K2, B55, and B30 were obtained with ion exchange chromatography, and their molecular weights were $2.41 \times 10^4$, $1.62 \times 10^4$, $6.42 \times 10^3$, $6.45 \times 10^3$, and $1.26 \times 10^4$ Da, found using gel permeation chromatography. The infrared spectrum results showed that these substances all contained polysaccharide characteristic absorption peaks, most of which contained O-H bonds, C-H bonds, hydroxyl and carboxyl bonds, and groups. The analysis of monosaccharide composition presented that EPS1a was composed of guluronic acid, rhamnose, and galactose, with a molar ratio of 2.7:1:2.4; EPS2b and EPS3c were composed of guluronic acid; EPS4d was composed of guluronic acid, glucose, and galactose, with a molar ratio of 1:1.1:1.2; and EPS5e was composed of glucose and galactose, with a molar ratio of 1.6:1, indicating the differences in the composition and structure of EPS produced by various strains. Compared with the control group that only had the starter added, adding EPS-producing strains could promote acid production and improve the texture of fermented goat milk, and its acidity, hardness, consistency, viscosity, and viscosity index were higher.

**Keywords:** extracellular polysaccharide; lactobacillus; purification and identification; fermented goat milk

## 1. Introduction

Lactic acid bacteria (LAB) are a kind of important probiotics, which are widely used in food because of their safety characteristics [1]. LAB are used by people around the globe to enhance the nutritional value and storage quality of perishable food in the early stage, and LAB are most widely used as a dairy starter for cheese, yogurt, buttermilk, sour cream, and other fermented dairy products. LAB can adhere to the gastrointestinal mucosa and produce inhibitory substances, which inhibit the growth of harmful microorganisms. They maintain a healthy balance with potentially harmful microorganisms by competitively excluding or producing organic acids, enzymes, extracellular polysaccharides, and antimicrobial compounds. They also have a variety of functions, such as reducing cholesterol, antioxidation, preventing colon cancer, and improving the flavor, texture, and nutritional properties of products [2–4].

Extracellular polysaccharide (EPS) is a carbohydrate secreted by LAB in the process of growth and metabolism. EPS is a water-soluble colloid with special physical properties.

The huge structural diversity and monosaccharide composition among EPS mediate the properties and functions of a large number of EPS types [5]. EPS from LAB, as a natural thickener, imparts food (such as yogurt) with beneficial rheological functional properties, imparts the product with proper viscosity, and reduces dehydration shrinkage [6].

Goat milk has more nutritional advantages than milk. Compared with milk, goat milk contains higher levels of unsaturated fatty acids, as well as total fat, vitamins, calcium, carbohydrates, and protein [7,8]. Goat milk, in particular, has greater fat and protein digestibility, as well as higher vitamin A, vitamin B, and calcium content [9]. The total composition of goat milk is higher than that of milk, but the lactose content is lower and the fat globules are smaller. The protein in goat milk is fragile during acidification in the stomach, which is more conducive to digestion than milk protein [10]. Goat milk is also used to treat different problems, including gastrointestinal disorders, diarrhea, constipation, etc. It also has health care functions such as antiallergy properties and improving immunity [9,11,12]. Although goat milk is better for health than milk, the consistency and viscosity of goat milk yoghurt are low and the texture is poor, reducing the overall acceptability for consumers [13]. The quality and texture of yogurt are also affected by the problems of low viscosity, gel rupture, or dehydration shrinkage in the manufacturing process of yogurt.

EPS can prevent dehydration shrinkage, improve the stability of the product by increasing the viscosity and elasticity of the product, and combine water or other milk ingredients to enhance the rigidity of the casein network. EPS and casein complex's viscoelasticity, water-holding ability, and particle size have all grown [14,15]. The EPS-producing strains could affect the rheological properties of dairy products, and the fermentation effects of each strain are different because of the structural characteristics of EPS [16]. EPS activity is related to its structural characteristics, such as glycosidic bond type, molecular weight, monosaccharide composition, and its connection type [17]. Therefore, in this study, EPS produced by five lactic acid bacteria (*Limosilactobacillus fermentum* B55, *Limosilactobacillus fermentum* B62, *Lactiplantibacillus plantarum* 7830, *Pediococcus acidilactici* B30, and *Lactobacillus helveticus* K2) was purified and identified. The five LAB with EPS-producing potential were used as goat milk starters to research their effects on the pH, acidity, hardness, consistency, viscosity, and viscosity index of fermented goat milk, so as to solve the above problems and provide a reference and technical support for improving the quality of this milk. The five kinds of LAB with high EPS yield were screened and identified out of 66 strains isolated from commercial dairy products, Kefir grains, and fermented bovine milk [18].

## 2. Materials and Methods

### 2.1. Preparation of EPS

Five LAB were provided by the School of Food Science and Engineering, Shaanxi University of Science & Technology.

The strains were activated for three generations at 37 °C for 24 h. Then, a total of 2% (*v/v*) each of five strains was inoculated into commercial MRS medium (37 °C, 48 h) to prepare the EPS, respectively. The EPS was obtained via centrifugation and dialysis using the method of Savadogo et al. [19]. Briefly, the first step was centrifugation (4 °C, 8000 rpm/min, 15 min) to remove the bacteria. In the second step, the supernatant was centrifuged with 10% trichloroacetic acid to remove the protein. In the third step, a double volume of ethanol was added (refrigerated at 4 °C overnight) and centrifuged, and the collected precipitates were dissolved in distilled water and placed in a dialysis bag (24 h).

### 2.2. Isolation and Purification of EPS from LAB

The EPS was purified using DEAE cellulose 52 ion exchange chromatography and Sepharose CL-6B cross-linked agarose gel filtration chromatography according to the method of Liu et al. [20]. Briefly, 0.03 g of EPS was weighed and dissolved in 6 mL of distilled water. Then, at a flow rate of 1.5 mL/min, gradient elution was carried out

using distilled water and solutions of 0.05, 0.1, and 0.3 mol/L NaCl. An amount of 5 mL was collected in each tube. Protein and nucleic acid impurities were detected with a UV-visible spectrophotometer. The EPS content was measured with the phenol–sulfuric acid method [21], and the DEAE–cellulose elution curve was produced with the horizontal coordinate of the number of eluting tubes and the vertical coordinate of the polysaccharide concentration. The eluted and collected EPS of LAB was further purified. NaCl eluent at a concentration of 0.1 mol/L was used at the flow rate of 0.5 mL/min to detect the EPS content of LAB in the eluent, and the elution EPS curve of the Sepharose CL-6B crosslinked agarose gel column was drawn.

### 2.3. Determination of Molecular Weight Distribution using Gel Permeation Chromatography (GPC)

A Waters515 pump laser detector and differential detector were used. The chromatographic column was an Ohpak series SB-802.5HQ and the mobile phase was NaNO3 (0.2 mol/L). The flow rate was set at 0.500 mL/min, the column temperature at 25 °C, and the sample size at 20 µL [22].

### 2.4. Determination of Monosaccharide Composition in Polysaccharides

Monosaccharides in the EPS were measured using the 1-phenyl-3-methyl-5-pyrazolone (PMP) derivation method [23]. Briefly, the EPS was hydrolyzed by 2.0 mol/L TFA (trifluoroacetic acid) for two hours at 100 °C in a sealed tube filled with nitrogen. Excess TFA was removed at 70 °C for drying by adding a small amount of methanol after hydrolysis and NaOH (0.3 mol/L) was added to fully dissolve the residue. Mixed monosaccharide standard solution or polysaccharide hydrolysate was added into a 5 mL test tube, and PMP methanol solution was added, swirled, and mixed. Next, it was placed in a 70 °C water bath for 2 h. To the mixture solution, 400 µL of HCl (0.3 mol/L) was added and chloroform was added for extraction. The aqueous phase was filtered through a 0.45 µm microporous membrane for high-performance liquid chromatography (HPLC) injection analysis. The chromatographic column was a C18 column (250 × 4.6 mm), mobile phase A was 0.1 mol/L sodium phosphate buffer (pH = 6.4), mobile phase B was acetonitrile, the detection wavelength was 250 nm, the column temperature was 30 °C, the flow rate was 1 mL/min, and the injection volume was 20 µL.

### 2.5. Structural Characteristics Determination of EPS using Infrared Spectroscopy

The German BRUKER ertex70 Fourier infrared spectrometer was used. The KBr tableting method (1 mg of EPS was added to 100 mg of KBr), a KBr beam splitter, a DigiTectTM detector system, a ROCKSOLIDTM interferometer with resolution of 0.4 cm$^{-1}$, scanning times of 105, and a test range of 4000~400 cm$^{-1}$ were used. The infrared spectrum of EPS was obtained using Prasad et al.'s approach, with some modifications [24].

### 2.6. Preparation of Fermented Goat Milk and Sensory Evaluation

Reconstituted goat milk (reconstituted milk with whole goat milk powder and water in a ratio of 1:8) was first prepared (6% sucrose was added), sterilized (90~95 °C, 10 min), and cooled to 45 °C. Then, it was inoculated (0.005% starter by mass ratio was added to the control group, and 0.0025% TW starter and 0.0025% freeze-dried LAB powder, respectively, were added to the experimental group), fermented at 43 °C for 4 h, ripened, and refrigerated (refrigerated at 0~4 °C overnight). Only the TW starter was added as the control. During the fermentation period of 1, 2, 3, and 4 h, the fermented goat milk samples were taken to determine pH, acidity, hardness, consistency, viscosity, and other indicators. TW was the control commercial starter, which was composed of *Lactobacillus bulgaricus* and *Streptococcus thermophilus*. The TW-fermented goat milk had good performance.

Five trained evaluators (2 males and 3 females, aged 20–45 years) performed sensory evaluations based on the methods of Chen He et al. [25] from our laboratory. The evaluators for describing and assessing the flavor of fermented goat milk had been trained according to the standard sensory analysis methods by ISO 6658:2017 [26]. Before the sensory evaluation,

the fermented goat milk products were placed in uniform transparent cups and labeled (with a random three-digit code). Color: 1 score (uniform color, milky white as the best); smell: 3 score (fresh scent, pure lactic acid flavor as the best); taste: 3 score (moderate sour and sweet, no astringency, no odor as the best); structural state: 3 score (uniform and delicate, good coagulability, no milk whey precipitation as the best).

### 2.7. Determination of PH Value of Fermented Goat Milk

A PHS-3C pH meter was used to measure the pH value of the fermented milk, and 3 parallels were made for each sample.

### 2.8. Determination of Acidity of Fermented Goat Milk

The acidity was determined using sodium hydroxide titration in Gilnier degrees (°T). The acidity of the fermented goat milk was tested using the method of Prasad et al. [27] and the national standard, with some modifications.

### 2.9. Determination of Texture (Hardness, Consistency, and Viscosity) of Fermented Goat Milk

The Texture Analyser (TA.XT plus) was applied in the TPA (texture profile analysis) mode with an A/BE (a back extrusion cell) probe, a back extension cell, and a probe pressure plate diameter of 20 mm. The probe ran at a rate of 1.0 mm/s before the test, at a rate of 1.0 mm/s during the test, at a rate of 10.0 mm/s after the test, and at a penetration test distance of 25 mm. The data acquisition rate was 400 pps, and the measurement temperature was 15 °C [28].

### 2.10. Statistical Analysis

The SPSS 25.0 statistical software was used to analyze the data (SPSS Inc., Chicago, IL, USA). The results were displayed as mean $\pm$ SD. Statistical analysis (Student's *t*-test) was performed and $p < 0.05$ was considered to be statistically significant.

## 3. Results and Discussion

### 3.1. EPS Isolation and Purification from Five Strains of LAB

The EPS of five strains of lactic acid bacteria was first separated and purified with a DEAE cellulose column, and then the components collected after purification were filtered and purified with a Sepharose CL-6B gel column. B62 extracellular polysaccharide was eluted in the DEAE cellulose column. The elution time of each component was different. Compared with the elution peaks, the component with the most EPS content was selected (tube No. 15–23) to mix and named EPS1 (Figure 1a). EPS1 was eluted with the Sepharose CL-6B gel column and a single peak was obtained, indicating a single component (Figure 1b). According to the elution peak position in the figure, tube numbers 10–30 of the eluent were combined (named EPS1a), and then they were freeze-dried in a vacuum to obtain white polysaccharide samples. The 7830 extracellular polysaccharide was eluted in a DEAE cellulose column and the content of mixed component was the highest (tube no. 88–95), which was named EPS2 (Figure 1c). The elution curve of EPS2 after elution with the Sepharose CL-6B gel column is shown in Figure 1d, indicating that this component was multicomponent in nature. According to the elution peak position in the figure, the elution solution of tubes 25–37 was combined (named EPS2b), and a white polysaccharide sample was obtained after vacuum freeze-drying. The K2 extracellular polysaccharide was eluted in the DEAE cellulose column, and the component with the largest EPS3 content, pipe No. 86–90, was selected for mixing, and named EPS3 (Figure 1e). EPS3 was eluted on the Sepharose CL-6B gel column, then combined with the eluent in tubes 30–37 according to the elution peak position, and then the white polysaccharide samples were obtained via vacuum freeze-drying and named EPS3c (Figure 1f). B55 extracellular polysaccharide was eluted in the DEAE cellulose column, and the component with the most content, pipe No. 32–39, was selected, mixed, and named EPS4 (Figure 2a). The elution curve of EPS4 after its elution on the Sepharose CL-6B gel column is shown in Figure 2b, indicating that

this component was multicomponent in nature. According to the position of the elution peak in the figure and the elution solution of tubes 20–30 in Figure 2b, after vacuum freeze-drying, it was named EPS4d, and the obtained polysaccharide sample was white. B30 extracellular polysaccharide was eluted in the DEAE cellulose column, and the component with the most content was mixed (tube no. 25–32), named EPS5, and purified continuously (Figure 2c). EPS5 was eluted on the Sepharose CL-6B gel column, and the elution curve is shown in Figure 2d, indicating that the component is multicomponent in nature.

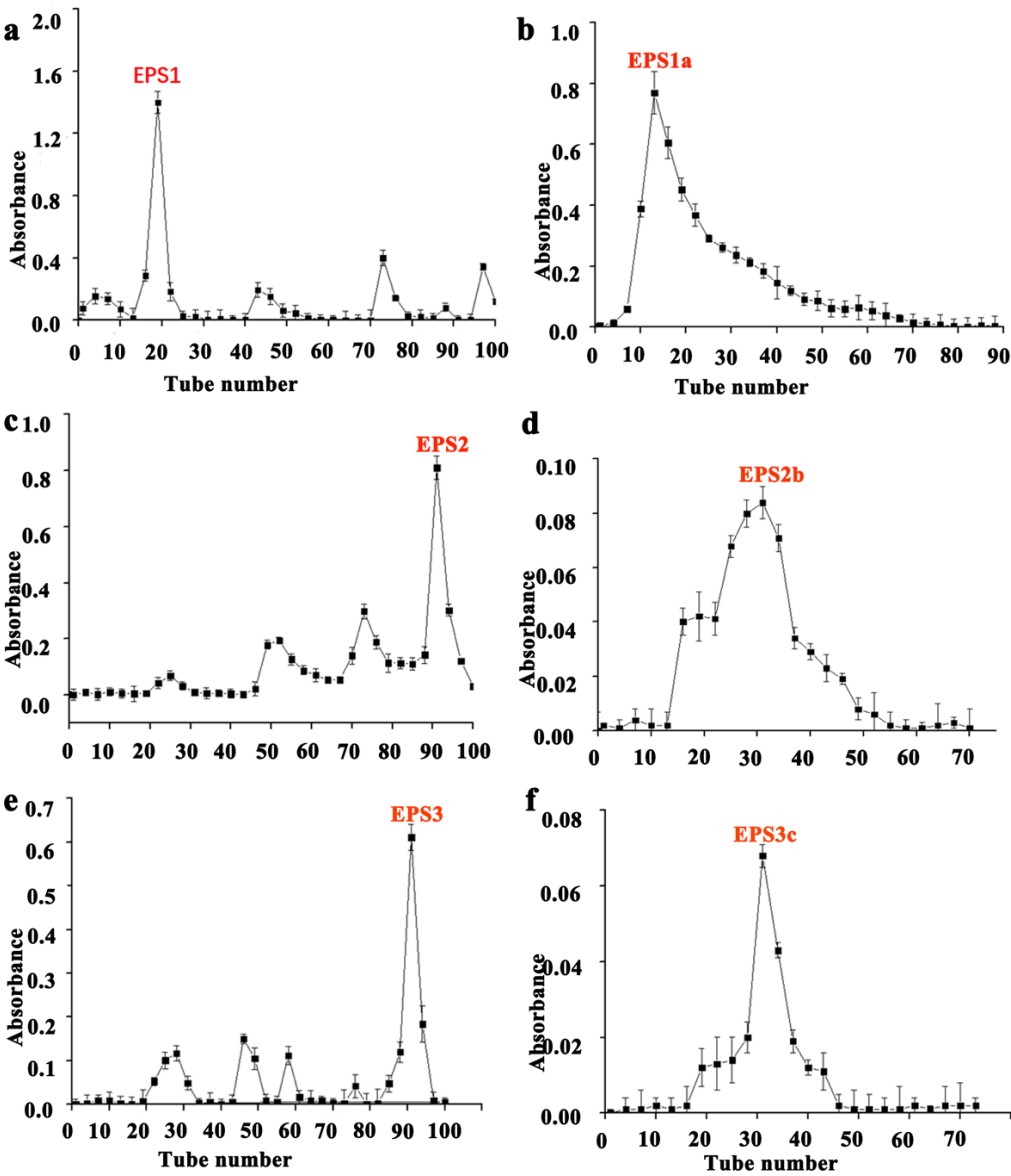

**Figure 1.** Isolation and purification of EPS from strains B62, 7830, and K2. (**a**) DEAE cellulose column elution curve of EPS of B62; (**b**) Sepharose CL-6B column elution curve of EPS1; (**c**) DEAE cellulose column elution curve of EPS of 7830; (**d**) Sepharose CL-6B column elution curve of EPS2; (**e**) DEAE cellulose column elution curve of K2; (**f**) Sepharose CL-6B column elution curve of EPS3.

According to the elution peak position in Figure 2d, the eluent of tubes 20–30, named EPS5e, was combined and freeze-dried in a vacuum to obtain a white polysaccharide sample, proving that the pigment had been removed.

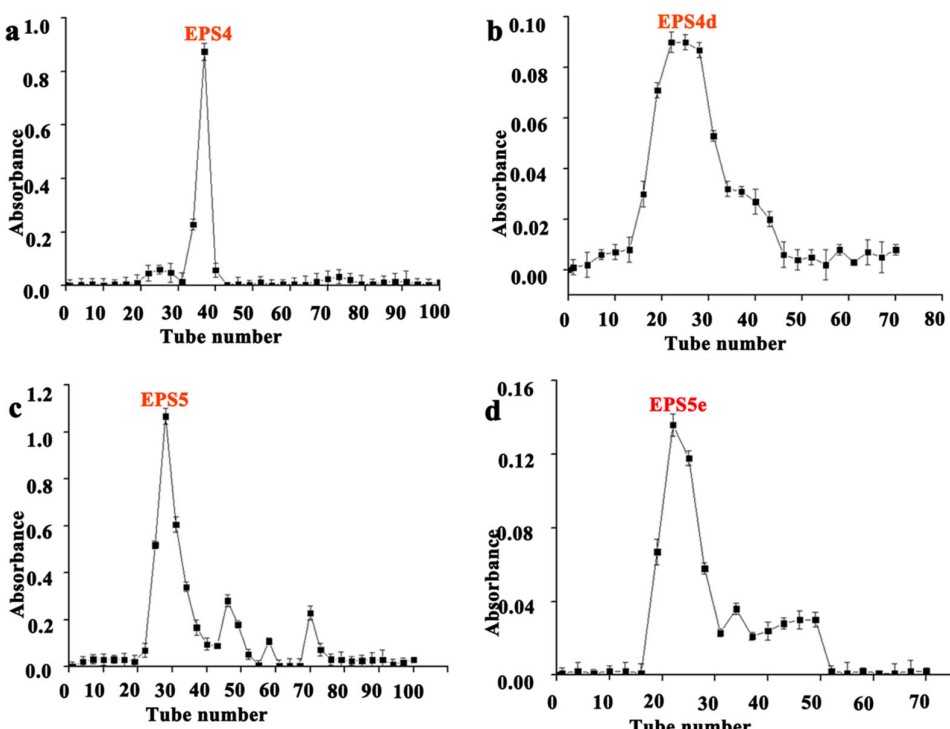

**Figure 2.** Isolation and purification of EPS from strains B55 and B30. (**a**) DEAE cellulose column elution curve of B55; (**b**) Sepharose CL-6B column elution curve of EPS4; (**c**) DEAE cellulose column elution curve of B30; (**d**) Sepharose CL-6B column elution curve of EPS5.

Our previous studies have reported that the EPS production of these five strains all exceeded 100 mg/L [18]. The EPS isolated by Wang et al. was a pale-yellow powder that was soluble in water at room temperature [29]. However, the extracellular polysaccharide isolated and purified in this study is a white powder. The pale-yellow color of the powder may be because a small amount of pigment remained in the process of separation and purification, or bacteria with different metabolites may have also had an effect. In this study, different methods from Mohamed et al. were used to separate and purify EPS, and the purpose was to further purify EPS and identify its chemical composition [30].

### 3.2. Molecular Weight Determination of EPS of Five Strains of LAB

The determination results of the molecular weight of EPS of five LAB strains are shown in Table 1. The standard curve was drawn according to the molecular weight of glucan and peak time, and the average molecular weight of the polysaccharide fraction was calculated from the sample retention time and the standard curve.

The results show that the relative weight average molecular weight (Mw) of EPS1a was $2.41 \times 10^4$ Da, with a number average molecular weight (Mn) of $8.16 \times 10^3$ Da and a dispersion degree (Mw/Mn) of 2.96. The molecular weight distribution was 3000.0~5000.0 g/mol accounting for 25.9%, 5000.0~40,000.0 g/mol accounting for 55.9%, 40,000.0~88,000.0 g/mol accounting for 13.5%, and 88,000.0~209,195.0 g/mol accounting for 4.7%. The Mw of EPS2b was $1.62 \times 10^4$ Da, the Mn was $9.36 \times 10^3$ Da, and the dispersion degree (Mw/Mn) was 1.735. The molecular weight distribution was 3000.0~5000.0 g/mol taking up 16.9%, 5000.0~11,000.0 g/mol taking up 29.4%, 11,000.0~48,000.0 g/mol taking up 49.9%, and 48,000.0~78,852.0 g/mol taking up 3.7%. The Mw of EPS3c was $6.42 \times 10^3$ Da, the Mn was $3.52 \times 10^3$ Da, and the dispersion degree (Mw/Mn) was 1.822. The molecular weight distribution was 1600.0~4300.0 g/mol making up 49.8%, 4300.0~7200.0 g/mol making up 17.0%, 7200.0~20,000.0 g/mol making up 26.7%, and 20,000.0~37,352.0 g/mol making up 6.5%. The Mw of EPS4d was $6.45 \times 10^3$ Da, the Mn was $4.22 \times 10^3$ Da, and the dispersion degree (Mw/Mn) was 1.531. The molecular weight distribution was 2200.0~3200.0 g/mol account-

ing for 29.1%, 3200.0~6200.0 g/mol accounting for 34.6%, 6200.0~8200.0 g/mol accounting for 14.3%, 8200.0~20,000.0 g/mol accounting for 18.5%, and 20,000.0~45,402.0 g/mol accounting for 3.5%. The Mw of EPS5e was $1.26 \times 10^4$ Da, the Mn was $4.87 \times 10^3$ Da, and the dispersion degree (Mw/Mn) was 2.582. The molecular weight distribution was 2000.0~3500.0 g/mol accounting for 39.2%, 3500.0~16,000.0 g/mol accounting for 32.9%, 16,000.0~28,000.0 g/mol accounting for 21.8%, 28,000.0~67,000.0 g/mol accounting for 3.1%, and 67,000.0~16,4802.0 g/mol accounting for 3.0%.

**Table 1.** Molecular weight of exopolysaccharides produced by five LAB.

| EPS | Mw (Da) | Mn (Da) | Mw/Mn | Molar Mass Distributions |
|---|---|---|---|---|
| EPS1a | $2.41 \times 10^4$ | $8.16 \times 10^3$ | 2.96 | 3000.0~5000.0 g/mol accounting for 25.9%, 5000.0~40,000.0 g/mol accounting for 55.9%, 40,000.0~88,000.0 g/mol accounting for 13.5%, and 88,000.0~209,195.0 g/mol accounting for 4.7%. |
| EPS2b | $1.62 \times 10^4$ | $9.36 \times 10^3$ | 1.735 | 3000.0~5000.0 g/mol accounting for 16.9%, 5000.0~11,000.0 g/mol accounting for 29.4%, 11,000.0~48,000.0 g/mol accounting for 49.9%, and 48,000.0~78,852.0 g/mol accounting for 3.7%. |
| EPS3c | $6.42 \times 10^3$ | $3.52 \times 10^3$ | 1.822 | 1600.0~4300.0 g/mol accounting for 49.8%, 4300.0~7200.0 g/mol accounting for 17.0%, 7200.0~20,000.0 g/mol accounting for 26.7%, and 20,000.0~37,352.0 g/mol accounting for 6.5%. |
| EPS4d | $6.45 \times 10^3$ | $4.22 \times 10^3$ | 1.531 | 2200.0~3200.0 g/mol accounting for 29.1%, 3200.0~6200.0 g/mol accounting for 34.6%, 6200.0~8200.0 g/mol accounting for 14.3%, 8200.0~20,000.0 g/mol accounting for 18.5%, and 20,000.0~45,402.0 g/mol accounting for 3.5%. |
| EPS5e | $1.26 \times 10^4$ | $4.87 \times 10^3$ | 2.582 | 2000.0~3500.0 g/mol accounting for 39.2%, 3500.0~16,000.0 g/mol accounting for 32.9%, 16,000.0~28,000.0 g/mol accounting for 21.8%, 28,000.0~67,000.0 g/mol accounting for 3.1%, and 67,000.0~164,802.0 g/mol accounting for 3.0%. |

Mw stands for weight average molecular weight; Mn stands for number-average molecular weight; and Mw/Mn stands for polydispersity.

There were significant differences in the distribution of Mw, Mn, and molecular weight of EPS produced by different LAB, which might be associated with the different characteristics of several LAB. The chemical structures and characteristics of EPS are diverse. Aside from yield, the molecular weight and monomer makeup of polymers have a key influence in defining EPS's viscosity and applicability. In the study by Wang et al., HPSEC-MALLS was used to measure the Mw and Mn of EPS. The Mw and Mn were $3.246 \times 10^4$ and $2.012 \times 10^4$ g/mol, respectively [29]. The Mw and Mn of EPS isolated and purified by Samar et al. were $4.9 \times 10^4$ and $5.4 \times 10^4$ g/mol, respectively [31]. This might be mainly associated with the type of sample, mass concentration, and column efficiency of the gel column. Jiang et al. detected the molecular weights of two EPSs (LPB8-0 and LPB8-1) isolated and purified from *L. pentosus* B8 with an HPSEC–RI–MALLS system, and the results showed that their molecular weights were $1.12 \times 10^4$ and $1.78 \times 10^5$ Da, respectively [32].

Meanwhile, the molecular weight of the EPS isolated from *Pediococcus pentosaceus* using gel permeation chromatography (equipped with an Agilent PL aquagel–OH MIXED–H column) was $7.6 \times 10^4$ Da [33]. In addition, the gel permeation chromatograph equipped with a TSK G-5000 PWXL column was used to measure the Mw ($6.61 \times 10^4$ Da) of the EPS from *L. plantarum* WLPL04 [34].

### 3.3. Analysis of Monosaccharide Composition of EPS Produced by Five Strains of LAB

The HPLC determination results of mixed standard monosaccharide derivatives are shown in Figure 3a, with clear peaks and significant separation. There are three peaks in the liquid chromatography results of EPS1a hydrolysate derivatives. According to the comparison between the retention times of EPS1a derivatives and standard sample derivatives, it can be seen that the retention time of EPS1a derivatives is basically the same as that of guluronic acid, rhamnose, and galactose. It can be inferred that guluronic acid, rhamnose, and galactose are components of EPS1a, and their molar ratio is 2.7:1:2.4 (Figure 3b). The liquid chromatography results of EPS2b hydrolysate derivatives show only one peak. EPS2b consists of a monosaccharide derivative of guluronic acid (Figure 3c). Similarly, EPS3c is composed of a monosaccharide derivative of guluronic acid (Figure 3d). The liquid chromatography results of EPS4d hydrolysate derivatives show three peaks. The retention time of EPS4d derivatives is basically the same as that of guluronic acid, glucose, and galactose. EPS4d is a heteropolysaccharide composed of the above three monosaccharides, with a molar ratio of 1:1.1:1.2 (Figure 3e). There are two peaks in the liquid chromatography results of EPS5e hydrolysate derivatives. The retention time of EPS5e derivatives is basically the same as that of glucose and galactose. It can be inferred that EPS5e is a heteropolysaccharide composed of these two monosaccharides, with a molar ratio of 1.6:1 (Figure 3f).

After hydrolysis and derivatization, the retention time of different EPSs purified using HPLC is different, indicating that the monosaccharide composition and molar ratio of EPS produced by different bacteria are different. The EPS isolated in Wang et al.'s investigation was predominantly composed of five different monosaccharides: D-galacturonic acid, glucose, rhamnose, D-galactose, and D-mannose in the ratio of 0.45: 1.0:3.02: 0.95: 3.25 [29]. The EPSR5 fraction is made up of glucose, xylose, arabinose, and galacturonic acid in the following proportions: 2.0:1.0:0.25:0.5, respectively, according to Samar et al.'s analysis of EPS [31]. The EPS produced by *Lactiplantibacillus plantarum* purified by Wang et al. consisted of arabinose, mannose, glucose, and galactose, with a molar ratio of 0.95:12.94:7.26:3 [6].

Some literature sources report that the EPS produced by *Lactiplantibacillus plantarum* is a heteropolysaccharide composed of glucose, galactose, mannose, arabinose, and other monosaccharides [35–37]. In our study, the EPS2b produced by *Lactiplantibacillus plantarum* was composed of guluronic acid. Our research indicated that the monosaccharide composition of EPS produced by five different strains was also different, suggesting that the characteristics of EPS produced by different strains are quite different. Therefore, it is necessary to further explore its role in improving the hardness and consistency of food and its application in fermented milk.

Our study demonstrated that there were some differences in the composition and properties of EPS produced by five strains of LAB. The production of bacterial EPS is controlled by genetic factors and regulated by environmental factors. Different strains have different genetic compositions and epigenetic metabolism, so the composition and structure of EPS produced by different strains of the same species may be different. U. Pachekrepapol et al. studied the detailed characteristics of EPS produced by seven strains of *Streptococcus thermoophilus* and found that the structure and composition were not completely consistent [38]. Meanwhile, the production of EPS is also related to the bacterial culture environment. Andrea et al. tested the ability of three different LAB to produce EPS with five different monosaccharides and disaccharides as the only carbon sources. The results showed that the EPS yield and composition of different strains cultured under different conditions had certain differences [39]. In addition, Xu et al. comprehensively summa-

rized the structure and function of EPS production of different LAB and *bifidobacterium* (1999–2018), and these studies showed that the chemical structure, technological characteristics, and physiological function of EPS production of different strains were different [40].

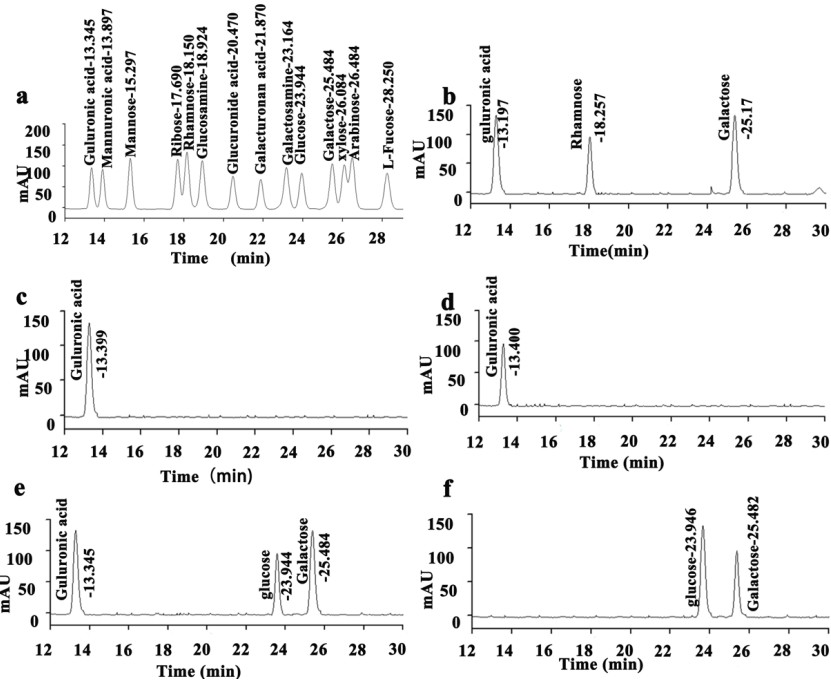

**Figure 3.** Determination of monosaccharide composition of EPS. (**a**) HPLC of mixed standard monosaccharide derivatives, (**b**) EPS1a derivatives, (**c**) EPS2b derivatives, (**d**) EPS3c derivatives, (**e**) EPS4d derivatives, and (**f**) EPS5e derivatives.

### 3.4. Infrared Spectrum Analysis of EPS Produced by Five Strains of LAB

Figure 4a indicates that EPS1a has a broad and strong absorption which appears at 3421.6 cm$^{-1}$, which is the performance of the O-H bond stretching vibration between sugar molecules or within molecules [41]. The characteristic absorption peak induced by the stretching vibration of the C-H bond is observable at 2962.57 cm$^{-1}$ [42]. The peak around 1200~950 cm$^{-1}$ is considered the fingerprint area of EPS. 1570.01 cm$^{-1}$ is the carboxyl asymmetric stretching vibration peak, and the peak at 1419.56 cm$^{-1}$ is the C-H bond variable angle vibration peak [41,43,44]. The two peaks of 1261.4 and 1083.96 cm$^{-1}$ are induced by two C-O stretching vibrations. The former is the C-O-C of the pyranose ether bond, while the latter is the hydroxyl absorption peak [45,46]. The absorption peak at 869.87 cm$^{-1}$ suggests that the end group carbon is the result of the joint action of α- and β-. Figure 4b illustrates that EPS2b has a wide and significant absorption peak centered at 3390.76 cm$^{-1}$, which is the performance of the stretching vibration of the O-H bond [41]. The characteristic absorption peak induced by the stretching vibration of the C-H bond is observable at 2967.85 cm$^{-1}$ [42]. At 1660.3 cm$^{-1}$, it is the asymmetric stretching vibration peak of the carboxyl group [29]. At 1450.2 cm$^{-1}$, there is C-H bond angular vibration [41,43,44]. The peaks at 1262.54 and 1087.52 cm$^{-1}$ are caused by two kinds of C-O stretching vibration, which are the C-O-C of pyranose internal ether bond and the hydroxyl absorption peak [45,46]. The peaks of 880.84 and 842.30 cm$^{-1}$ terminal carbon are the result of the joint action of α- and β-. Figure 4c shows that EPS3c has a broad and strong absorption peak at 3394.61 cm$^{-1}$, which is the performance of the stretching vibration of the O-H [41]. There is absorption which appears at 2966.43 cm$^{-1}$, which is the characteristic absorption caused by the stretching vibration of the C-H bond [42]. At 1660.66 cm$^{-1}$, it is the asymmetric stretching vibration peak of the carboxyl group [29], and at 1406.06 cm$^{-1}$, it is the variable angle vibration of the C-H bond [41,43,44]. The peaks at 1263.34 and 1097.46 cm$^{-1}$ are induced by two kinds of C-O stretching vibration. One

is the C-O-C of the pyranose inner ether bond, and the other is the hydroxyl absorption peak [45,46]. The peaks at 881.44, 840.93, and 676.99 cm$^{-1}$ prove that the end group carbon is the result of the joint action of α- and β-. Figure 4d demonstrates that EPS4d has a wide and significant absorption peak centered at 3421.62 cm$^{-1}$, which is the performance of the stretching vibration of the O-H bond [41]. There is an absorption peak at 2962.57 cm$^{-1}$, which is the characteristic absorption peak caused by the C-H bond stretching vibration [42]. The asymmetric stretching vibration peak of the carboxyl group is at 1654.87 cm$^{-1}$ [29], and the variable angle vibration of the C-H bond is at 1392.56 cm$^{-1}$ [41,43,44]. The peaks at 1265.26 and 1080.11 cm$^{-1}$ are induced by two C-O stretching vibrations. One is the C-O-C of the pyranose inner ether bond, and the other is the hydroxyl absorption peak [45,46]. The peak at 840.93 cm$^{-1}$ proves that the terminal carbon is the result of α-type action, and 638.41 cm$^{-1}$ is the characteristic absorption peak of glucose residues. Figure 4e shows that EPS5e has a wide and significant absorption peak at 3400.41 cm$^{-1}$, which is the performance of the O-H bond stretching vibration [41]. The absorption peak centered at 2970.28 cm$^{-1}$ is the characteristic absorption caused by the stretching vibration of the C-H bond [42], 1662.59 cm$^{-1}$ is the asymmetric stretching vibration of the carboxyl group [29], and 1407.99 cm$^{-1}$ is the variable angle vibration of the C-H bond [41,43,44]. The two peaks at 1265.26 and 1095.53 cm$^{-1}$ are caused by two kinds of C-O stretching vibration. One is the C-O-C of the pyranose inner ether bond, and the other is the hydroxyl absorption peak [45,46]. The peak at 842.86 cm$^{-1}$ shows that the end group carbon is the result of α-type action, and the peak centered at 632.63 cm$^{-1}$ is the characteristic absorption of glucose residues.

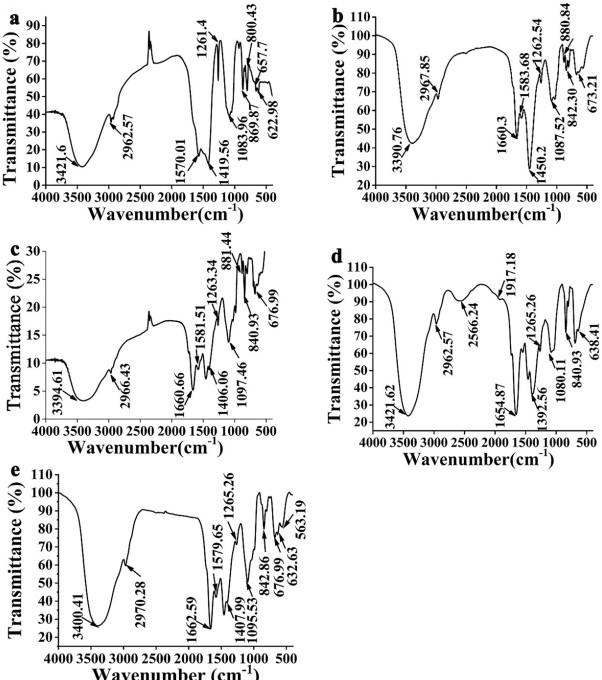

**Figure 4.** Infrared absorption spectrum of EPS of (**a**) EPS1a, (**b**) EPS2b, (**c**) EPS3c, (**d**) EPS4d, and (**e**) EPS5e.

The bonds and groups corresponding to EPS signals at different peaks have been demonstrated in a large body of literature [35–37,47]. The FT-IR (Fourier transformation infrared) results are consistent with the purified EPS's physicochemical properties. In a word, the FT-IR analysis shows that EPS from different strains has structural differences. In the experiment, the peak times of the characteristic absorption peaks of glucose residues with EPS4d and EPS5e were different. The reason for the difference may be that the absorption frequency of the group shifted to the direction of low wavenumber after the hydrogen containing the hydrogen group was replaced by deuterium during the EPS

reaction, and the induction effect between covalent bonds reduced the absorption intensity of the polar group. Then, the conjugation effect weakened the double-bond effect in the original group and reduced the force constant, which together caused the red shift phenomenon of the infrared spectrum.

*3.5. Evaluation of the Effect of Five Strains of EPS-Producing LAB on the Texture of Fermented Goat Milk*

Only the TW starter was added as the control. During the fermentation period of 1, 2, 3, and 4 h, the fermented goat milk samples were taken to measure pH, acidity, texture, and other indicators. TW is the control commercial starter, which is composed of Lactobacillus bulgaricus and Streptococcus thermophilus. The pH of the fermented goat milk decreased with time, while the acidity increased over time. The pH value of the fermented goat milk with five added EPS-producing strains decreased rapidly, and the pH values at the end of fermentation were 4.8 for the TW starter, 4.3 for the TW starter +B62, 4.21 for the TW starter +7830, 4.15 for the TW starter +K2, 4.42 for the TW starter +B30, and 4.21 for the TW starter +B55 (Figure 5a). The acidity of the fermented goat milk with five added strains of EPS-producing LAB also increased faster than that with only TW starter added. At the end of fermentation, the acidity values at 4 h were 80.44 °T for the TW starter, 105.33 °T for the TW starter +B62, 99.84 °T for the TW starter +7830, 103.33 °T for the TW starter +K2, 107.03 °T for the TW starter +B30, and 98.62 °T for the TW starter +B55 (Figure 5b). With the extension of time, the hardness and consistency of the fermented goat milk increased. During the fermentation process of LAB, lactose in the goat milk is decomposed into lactic acid, which makes the protein in the milk denature and coagulate to form a fermented yoghurt with viscosity. The hardness and consistency of fermented goat milk with added EPS-producing LAB are higher than those of the pure starter, and the hardness values at the end of fermentation were 10.03 g for the TW starter, 11.68 g for the TW starter +B62, 11.43 g for the TW starter +7830, 11.20 g for the TW starter +K2, 10.72 g for the TW starter +B30, and 11.53 g for the TW starter +B55, respectively. The consistency values at the fermentation endpoint of 4 h were 263.78 g·s for the TW starter, 287.91 g·s for the TW starter +B62, 293.66 g·s for the TW starter +7830, 298.51 g·s for the TW starter +K2, 293.51 g·s for the TW starter +B30, and 297.83g·s for the TW starter +B55 (Figure 5c,d). The interaction between the EPS and milk protein affected the texture properties of the fermented milk, especially the viscosity of the sour milk. During the fermentation process, the viscosity of the fermented goat milk increased with the extension of fermentation time and reached the maximum value at 4 h (Figure 5e,f). The variation trend of each strain was basically the same. The viscosity and viscosity index of the fermented goat milk of the five added strains of EPS-producing LAB were higher. The viscosity values at the fermentation endpoint of 4 h were: TW starter culture, 237.53 mPa·s; TW starter culture +B62, 284.92 mPa·s; TW starter culture +7830, 319.59 mPa·s; TW starter culture +K2, 341.13 mPa·s; TW starter culture +B30, 337.57 mPa·s; and TW starter culture +B55, 309.24 mPa·s. The viscosity indexes at the fermentation endpoint of 4 h were: TW starter culture, −9.56 g·s; TW starter culture +B62, −13.51 g·s; TW starter culture +7830, −12.67 g·s; TW starter culture +K2, −12.12 g·s; TW starter culture +B30, −11.44 g·s; and TW starter culture +B55, −13.30 g·s (Figure 5e,f). The results of sensory evaluation showed that the textures of the fermented goat milk were improved by adding five strains of EPS-producing LAB. TW + B62 had an extremely significant value ($p < 0.001$) and TW + K2 had a highly significant value ($p < 0.01$) compared with the control group (Figure 6).

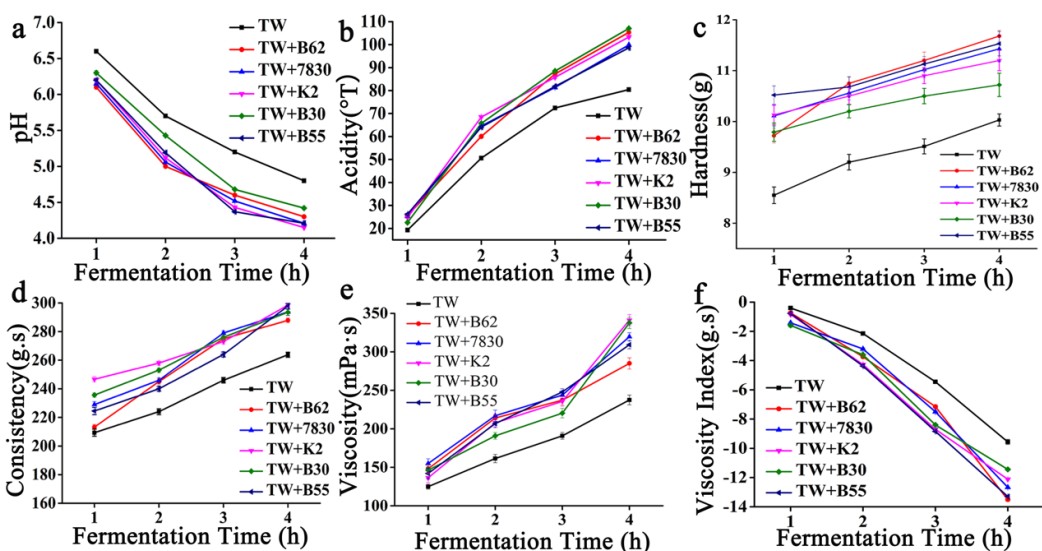

**Figure 5.** The effect of five strains of EPS-producing LAB on the quality of goat milk fermentation. (**a**) pH, (**b**) acidity, (**c**) hardness, (**d**) consistency, (**e**) viscosity, and (**f**) viscosity index.

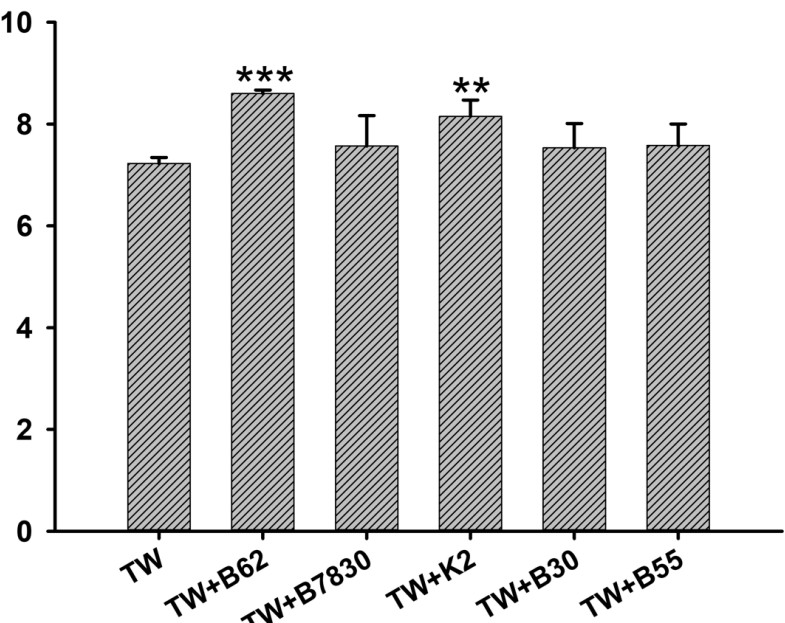

**Figure 6.** Sensory evaluation of goat yogurt fermented with five kinds of EPS-producing LAB. Values are expressed as mean ± SD ($n = 3$). **, and *** indicate $p < 0.05$, $p < 0.01$, and $p < 0.001$, respectively, compared with the TW control group.

Our study shows that the addition of EPS-producing LAB increases the hardness and consistency of fermented goat milk and significantly improves its quality. Due to the interaction with food components, microbial EPS has a functional role in food processing as it can enhance the sensory and rheological properties of food. EPS can be used as a plasticizer, stabilizer, thickener, or emulsifier [48]. Therefore, due to their high yield, EPS LAB are expected to become an excellent strain in the fermentation industry. Because of the structural characteristics of EPS, the monomer composition, bond type, molecular weight, and other factors can change the viscoelastic properties of food in different ways [49]. We analyzed the structural composition of five kinds of EPS-producing lactic acid bacteria, and the results show that LAB producing different EPS could bring different effects on the

quality of fermented goat milk. Xue et al.'s study found that, compared with commercial starter cultures, sour milk fermented with screened LAB with high-yield EPS had better texture and sensory properties, which were expected to reduce the application of food stabilizers [50]. Chengcheng et al. also obtained a higher apparent viscosity and better quality than commercial starter cultures by using EPS-producing LAB to ferment soybean milk [51]. The ability of LAB strains to produce EPS varies greatly in quantity and quality, which poses a limitation in the commercial application of LAB. However, our results show that EPS-producing LAB can significantly enhance the quality of fermented goat milk, which is expected to be further optimized for industrial application.

## 4. Conclusions

In this study, the EPS produced by five strains of LAB (Limosilactobacillus fermentum B55, Limosilactobacillus fermentum B62, Lactiplantibacillus plantarum 7830, Pediococcus acidilactici B30, and Lactobacillus helveticus K2) was purified and identified, and its effects on the texture and sensory properties of fermented goat milk were analyzed. The results show that the purified components of strains B62, 7830, K2, B55, and B30 EPS were obtained using ion exchange chromatography. Using gel permeation chromatography, the molecular weights of the purified EPS were determined to be $2.41 \times 10^4$, $1.62 \times 10^4$, $6.42 \times 10^3$, $6.45 \times 10^3$, and $1.26 \times 10^4$ Da, respectively. FT-IR analysis showed that the EPS of the different strains had structural differences. The monosaccharide composition analysis showed that the monosaccharide composition and molar mass ratio of EPS produced by different strains were different, which indicates that the composition and structure of EPS produced by different strains have significant differences. The application of EPS strains in goat milk fermentation shows that they can promote acid production and improve the texture of fermented goat milk, whose acidity, hardness, consistency, viscosity, and viscosity index were higher than those of the control group, which only had starter added to it. In addition, the sensory scores of the TW + B62 ($p < 0.001$) and TW + K2 ($p < 0.01$) groups were significantly higher than those of the TW group. In general, the five strains of EPS-producing LAB have good technological properties, which can significantly improve the texture and quality of fermented goat milk. We isolated, purified, and identified the structure of these five strains of EPS-producing LAB, which was meant to give a theoretical foundation for the industrial application and development of fermented goat milk.

**Author Contributions:** G.S. and L.C. designed the experiments. G.L., J.W., Z.W., Y.L. (Yu Liu) and Y.L. (Yilin Li) performed the experiments. G.L. prepared the manuscript. All authors have read and agreed to the published version of the manuscript.

**Funding:** The work was partially supported by the Scientific Research Program Funded by Shaanxi Provincial Education Department (No. 22JC022), Department-city linkage key projects sub-topic of Shaanxi Province (No. 2022GD-TSLD-58-2), Key R&D Projects in Shaanxi Province (No. 2023-YBNY-182 and 194), the key industrial chain "revealing the list and leading" project in Shaanxi province, the Shaanxi Province Qin Chuangyuan "scientist + engineer" team construction project (Program No. 2023KXJ-122), and the National Nature Science Foundation of China (32101908).

**Institutional Review Board Statement:** Not applicable.

**Informed Consent Statement:** Not applicable.

**Data Availability Statement:** Not applicable.

**Conflicts of Interest:** The authors declare no conflict of interest.

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
