# Peer review of "Purification and Identification of EPS Produced by Five Lactic Acid Bacteria and Evaluation of Their Effects on the Texture of Fermented Goat Milk"

_fermentation, doi:10.3390/fermentation9060527_

Round 1

Reviewer 1 Report

This submission reported how the five LAB strains affect the EPS production combined with goat milk fermentation. Meanwhile, texture of the fermented GM was analyzed. The following comments should be addressed before futher consideration.

1. The authors claimed that they isolated and identified the EPS produced from the 5 LAB samples. However, based from the results of Table 1, the EPSs produced from each strain were mixture instead of pure, single and identical EPS.

2. Line 25: We all know that LAB will produce lactic acid which will reduce pH and therefore change the protein solubility. What's the difference b/w your strains and control group?

3. A good discussion of why the 5 trains will produce different EPSs is needed.

4. A sensory evaluation is suggested since the results from Fig 5 would not be enough to represent the quality of fermented goat milk.

The English writing is easy to follow.

Author Response

Dear Reviewer

We are very appreciated that you read our manuscript carefully and provided many important suggestions for improving our work.  Based on these questions, we made substantial revisions to our manuscript and uploaded the reply file.

Reviewer 2 Report

This manuscript presents in detail the characteristics of five LAB fermentations to produce EPS and their practical application to goat milk fermentation. It is a reference value for future application of LAB in goat milk fermentation. However, there is no discussion on why these five LAB strains were chosen and the actual characteristics of the goat milk fermented by these five LAB strains, especially the sensory properties. It is suggested to add the reference for the selection of LAB in the paper, and to compare the advantages and disadvantages of the fermented goat milk characteristics and to identify which LAB strain is most suitable for fermentation of goat milk.

#1. In P2 Line71, total five different lactic acid bacteria were used in this manuscript, please explain the reasons for selecting these five strains of lactic acid bacteria, and provide supporting references.

#2. In P3 Line119, the type of sample used for IR scanning is not clearly described. Please add a description and clearly explain the pre-processing steps.

#3. In P4 Line187, it is showed that “Wang et al. was a pale-yellow powder”, but in this study is a white powder. Please provide a reasonable explanation for this section and provide supporting references.

#4. In P7 Line 234, it is showed that “This might be mainly associated with the type of sample, mass concentration, and column efficiency of gel column.”, however, the possible speculation of this phenomenon is not presented. Please supplement this part of the description and provide the corresponding references.

#4. In Figure 3, there are peaks of glucose in EPS4d and EPS 5e. But in the Figure 4, at 638.41 cm-1 is the characteristic absorption peak of glucose residues for EPS4d and at 632.63 cm-1 is the characteristic absorption of glucose residues for EPS5e. The same is the glucose residue, but the peak time are different, please explain. The use of FT-IR spectroscopy for the analysis of mixtures is already susceptible to the interference of other compounds. Therefore, the use of FT-IR absorbance spectroscopy to investigate the composition of compounds in this study seems to be inappropriate. Please explain the reasons for this analysis. If only to prove the result in P10 Line 331 “FT-IR analysis shows that EPS from different strains has structural differences.” Then should you consider whether it is necessary to describe the absorption phenomena of so many different functional groups in this paragraph without providing further explanation.

Please provide the ,oderate editing of English language.

Author Response

Dear reviewer

We are very appreciated that you read our manuscript carefully and provided many important suggestions for improving our work. Based on these questions, we have revised our manuscript substantially as following below.

#1. In P2 Line71, total five different lactic acid bacteria were used in this manuscript, please explain the reasons for selecting these five strains of lactic acid bacteria, and provide supporting references.

Response 1: Thank you very much for your professional suggestions. We added the relevant information about five kinds of LAB in lines 84-86 of the manuscript. The emendation is as follow:

The five kinds of LAB with high EPS yield were screened and identified out from 66 strains isolated from commercial dairy products, Kefir grains and fermented bovine milk [1].

#2. In P3 Line119, the type of sample used for IR scanning is not clearly described. Please add a description and clearly explain the pre-processing steps.

Response 2: Thank you very much for your professional suggestions. We added the description of the pre-processing steps in line 138-139 of our manuscript. The emendation is as follow:

The German BRUKER ertex70 Fourier infrared spectrometer was used. KBr tableting method (1mg EPS was added to 100mg KBr), KBr beam splitter, DigiTectTM detector system, ROCKSOLIDTM interferometer, with resolution of 0.4 cm-1, scanning times of 105, and test range of 4000~400 cm-1. The infrared spectrum of EPS was obtained using Prasad et al.'s approach, with some modifications.

#3. In P4 Line187, it is showed that “Wang et al. was a pale-yellow powder”, but in this study is a white powder. Please provide a reasonable explanation for this section and provide supporting references.

Response 3: The pale-yellow powder may be that a small amount of pigment remains in the process of separation and purification, or bacteria with different metabolites may also have an effect.

#4. In P7 Line 234, it is showed that “This might be mainly associated with the type of sample, mass concentration, and column efficiency of gel column. ”, however, the possible speculation of this phenomenon is not presented. Please supplement this part of the description and provide the corresponding references.

Response 4: This part of the description may not be very accurate, it has been removed in line 274 of our manuscript.

#5. In Figure 3, there are peaks of glucose in EPS4d and EPS 5e. But in the Figure 4, at 638.41 cm-1 is the characteristic absorption peak of glucose residues for EPS4d and at 632.63 cm-1 is the characteristic absorption of glucose residues for EPS5e. The same is the glucose residue, but the peak time are different, please explain. The use of FT-IR spectroscopy for the analysis of mixtures is already susceptible to the interference of other compound

Response 5: Thank you for your kind reminder. In the experiment, the absorption frequency of the group shifts to the direction of low wavenumber after the hydrogen containing the hydrogen group is replaced by deuterium during the EPS reaction, and the induction effect between covalent bonds reduces the absorption intensity of the polar group. Then, the conjugation effect weakens the double bond effect in the original group and reduces the force constant, which together cause the red shift phenomenon of the infrared spectrum.

Thank you for all your valuable comments!

Yours sincerely,

12, 5, 2023

References

[1]Shangguan, W., Chen, H., Li, Y., Wang, Z., Guo, H., & Meng, J. (2019). Screening and identification of new types of exopolysaccharides-producing lactic acid in the inner mongolia dairy products. Acta Universitatis Cibiniensis. Series E: Food Technology23(2), 75-84.

Reviewer 3 Report

Dear authors,

 I am ready with the critical review of the manuscript  Purification and Identification of EPS Produced by Five Lactic 2 Acid Bacteria and Evaluation of Their Effects on the Texture of 3 Fermented Goat Milk“.

Title is correct and correspond well to the data presented.

The overall strategy of the study is correct. However, presentation did not allow to estimate original elements of the work.

Extensive editing of  Materials and methods is  absolutely required :

1.       There is no detailed data about the LAB strains used as producers. Only was mention their origin. There is no information on their species identification (collection? or isolates). If some data about the strains were published, it’s necessary to be presented. Data on cultivation conditions for obtaining EPS are also missing. The MRS broth used is commercial or modified?

Presented data is too limited: Page 2, line 81: Five LAB were provided by the School of Food Science and Engineering, Shaanxi 80 University of Science & Technology. 81

2. To my opinion presented Preparation of crude EPS“ is inaccurate and unclear. Thus, any possibility for following purification step cannot be estimated by the writers.

Ex. Page 2. Line 79: Preparation of crude EPS: The strains were activated for three generations at 37 °C for 24 h. Then, A total 2 vol 82 % each of five strains were inoculated into MRS medium (37 °C, 48 h) to prepare EPS, 83 respectively. The crude EPS was obtained by centrifugation and dialysis using the method 84 of Savadogo et al. [18].

3. Please be so kind to explain correctly partPreparation of fermented goat milk. The term raw milk in specialised literature is usually used for whole unpasteurised/non-sterile milk. The authors use de facto reconstituted dry goat milk.

Ex. Page 3, lines 124 -125: Preparation of fermented goat milk

Raw milk (reconstituted milk with full fat goat milk powder and water in a ratio of 125 1:8) was first prepared (with extracted purified polysaccharide and sucrose added), sterilized (90~95 °C, 10 min), and cooled to 45 °C.  

 4. Point 2.9 “2.9. Determination of texture (hardness, consistency and viscosity) of fermented goat milk 141

TA.XT plus physical property tester was applied with TPA mode, A/BE probe, back 142 extension cell and probe pressure plate diameter of 20 mm. The probe run at a rate of 1.0 143 mm/s before test” is unclear done.

Any abbreviation used must be presented in full the first time it is used

 5. The work would benefit from an accurate presentation of the model for technological evaluation of fermented milks. Key details on the starter used with which the EPCs are combined are missing.  A comparative study of the effects of EPCs cannot be carried out without proven standardisation as % added or as composition. What about the duration of fermentation process? No data!

 6. Missing details for time points of acidity and pH measurement, predicted for EPS effects assessment.

 Results and discussion

·       Before to be purified the EPS have to be produced during the fermentation processes in MRS broth. Such data will be innovative and important in case of comparative study of five different strains belonging to different genus of LAB with different metabolism of the sugars. However, the results starts directly with purification. From this point of view is important to be presented the carbon used in MRS broth for production study.

·       In my humble judgment, Fig. 1 and Fig. 3 should be edited0 the font and size of the abscissa and ordinate labels are large and do not meet the requirements for formatting the results according to the Journal Instructions.

·       It would be good to comment on differences in milk coagulation times, especially if there are differences

·       Minor editing of English language will be appropriate:

Ex. Page 1 , line 37-38:  LAB can attach to the surface of the gastrointestinal  mucosa, producing inhibitory substances that inhibit the growth of harmful microorganisms.

·       Please avoid utilization of long sentences:

Ex. Page 1 , line 39-43:  They maintain a healthy balance with potentially harmful microorganisms by competitively excluding or producing organic acids, enzymes, extracellular polysaccharides, and antimicrobial compound, and have multiple functions, such as reducing cholesterol, anti-oxidation, preventing colon cancer, and improving the flavor, texture, and nutritional  properties of products.”  

In conclusion: Notwithstanding the remarks thus made, I believe that the authors may revise the submitted manuscript and recommend that it be resubmitted

Minor editing of English language will be appropriate:

Ex. Page 1 , line 37-38 LAB can attach to the surface of the gastrointestinal  mucosa, producing inhibitory substances that inhibit the growth of harmful microorganisms.

·       Please avoid utilization of long sentences:

Ex. Page 1 , line 39-43 They maintain a healthy balance with potentially harmful microorganisms by competitively excluding or producing organic acids, enzymes, extracellular polysaccharides, and antimicrobial compound, and have multiple functions, such as reducing cholesterol, anti-oxidation, preventing colon cancer, and improving the flavor, texture, and nutritional  properties of products.”

Latin names of microorganisms must be in  Italic. (Ex. name of starter used)

Author Response

Dear reviewer

We are very appreciated that you read our manuscript carefully and provided many important suggestions for improving our work.  Based on these questions, we made substantial revisions to our manuscript and uploaded the reply file.

Round 2

Reviewer 1 Report

I am happy with the revision.

Author Response

Thank you again for your work and splendid suggestions.

Reviewer 2 Report

For the response of the #3 and #5, please add them to the manuscript.

For the #4, please provide an explanation instead of deleting this paragraph.

For the P4 Line 143-147, the scale of sensory data is not clear enough. Why the score of color is 1, and the scores of smell, taste, and structural state are all 3. Please describe this section more clearly

Please provide the appropriate revise about the grammar of articles

Author Response

(The authors gave the same response as above.)

Reviewer 3 Report

Dear authors,

I am ready with the critical review of revised manuscript titled:“ Purification and Identification of EPS Produced by Five Lactic  Acid Bacteria and Evaluation of Their Effects on the Texture of  Fermented Goat Milk“.

I appreciate  your effort to edit the manuscript and to answer the questions. The missing details on Material and Methods were added.  Now the microbiological part of the experiments is presented and the missing details are added.

Now the fig 1 and 3 are very well .

The authors declare that two native English speakers are improved the language. However small technical mistakes (such  as intervals, letter, ) exist again and have to be avoid: ex.

·       Page 2 LINE 87 - Briefly, the first step was centrifugation (4°C, 8000 r/min, 15 min) – rpm/min or rcf what is correct? Please precise

·       Page 3 -135 line - experimental group were added 0.0025%TW starter and 0.0025%

Fig. 4. Also needs  to be edited, (font size of a.b.c.d.)  like figs 1  and 3

Author Response

(The authors gave the same response as above.)
